# Edible Insects as a Novel Source of Bioactive Peptides: A Systematic Review

**DOI:** 10.3390/foods12102026

**Published:** 2023-05-17

**Authors:** Carla S. S. Teixeira, Caterina Villa, Joana Costa, Isabel M. P. L. V. O. Ferreira, Isabel Mafra

**Affiliations:** REQUIMTE-LAQV, Faculdade de Farmácia, Universidade do Porto, Rua de Jorge Viterbo Ferreira, 228, 4050-313 Porto, Portugal; cteixeira@ff.up.pt (C.S.S.T.); cvilla@ff.up.pt (C.V.); jbcosta@ff.up.pt (J.C.); isabel.ferreira@ff.up.pt (I.M.P.L.V.O.F.)

**Keywords:** entomophagy, bioactive peptides, gastrointestinal digestion, health benefits, systematic review

## Abstract

The production of food and feed to meet the needs of the growing world’s population will soon become a serious challenge. In search for sustainable solutions, entomophagy is being proposed as an alternative source of proteins, with economic and environmental advantages when compared to meat. Edible insects are not only a valuable source of important nutrients, but their gastrointestinal digestion also originates small peptides with important bioactive properties. The present work intends to provide an exhaustive systematic review on research articles reporting bioactive peptides identified from edible insects, as demonstrated by *in silico*, *in vitro*, and/or *in vivo* assays. A total of 36 studies were identified following the PRISMA methodology, gathering 211 potentially bioactive peptides with antioxidant, antihypertensive, antidiabetic, antiobesity, anti-inflammatory, hypocholesterolemia, antimicrobial, anti-severe acute respiratory syndrome coronavirus type 2 (SARS-CoV-2), antithrombotic, and immunomodulatory properties, originated from the hydrolysates of 12 different insect species. From these candidates, the bioactive properties of 62 peptides were characterized in vitro and 3 peptides were validated in vivo. Data establishing the scientific basis of the health benefits associated with the consumption of edible insects can be a valuable contribution to overcoming the cultural issues that hinder the introduction of insects in the Western diet.

## 1. Introduction

According to the United Nations projections, the world’s population is expected to grow from the current 8 billion in 2022 to nearly 9.7 billion in 2050 (https://www.un.org/en/global-issues/population, accessed on 13 October 2022), which will demand a dramatic intensification of food and feed production. Additionally, the decrease of cultivation areas resulting from the climate changes and industrial development, together with the effects of the temperature changes on the crop yields, are serious challenges to overcome by the next generations. Currently, there is an increasing pursuit for sustainable solutions, which may rely on extending the ancient local food practices to a global scale, such as the case of entomophagy (practice of eating insects) that has been gaining relevance in recent years. Although this is presently considered as a new issue, the consumption of insects was already suggested as a source of food to fight the problem of food shortage decades ago by Meyer-Rochow [1]. This practice is designated as anthropo-entomophagy and constitutes the major source of nutrition as an alternative to animal and plant proteins. Presently, it is estimated that 2086 insect species are consumed as foods in 3071 ethnic groups from 130 countries [2]. However, the introduction of insects for human consumption in the Western countries, especially in Europe, is facing some resistance mainly due to cultural issues. To obviate the negative perception of insect consumption, some strategies are being adopted by the food industry, such as converting insects into flours that can be easily incorporated in processed foods, thus avoiding the repulsion caused by the insects’ physiognomy.

Insects possess an enormous biodiversity with more than 5.5 million species being identified so far [3], representing a large biomass. Their breeding has several environmental and economic advantages compared to the traditional protein sources (meat and plants), including (i) high reproduction rate, (ii) high feed conversion efficiency, (iii) small areas of rearing land required, avoiding deforestation, (iv) minor water needs, (v) lower emission of greenhouse gases and ammonia, (vi) lower economical investment in technology, and (vii) potential to reduce the use of insecticides when the collected insects are considered pests (e.g., desert locust) [4].

The nutritional composition of insects varies with the species, the development stage (larvae, pupae, adult), the diet, and the applied processing for its consumption (full insect, flour, baked, boiled, etc.). In general, they are excellent sources of protein and other important nutrients. Insects contain 13–81% of proteins (in dry matter) [4] that are composed of 46–96% of essential amino acids [5], having a digestibility between 76 and 96% [6]. Insects also have a significant content of fiber (8–27% in dry matter), mono- and polyunsaturated fatty acids (10–60% in dry matter), minerals (e.g., copper, zinc iron, manganese, magnesium, phosphorus, or selenium), and vitamins (e.g., A, B1–12, C, D, E, K) [4]. Additionally, insects are a good source of bioactive peptides (3–20 amino acids residues in length that promote beneficial effects for human health) [7], including antihypertensive [8], antidiabetic [9], antioxidant [10], antiobesity [11], immunomodulatory [12], anti-inflammatory [13], antimicrobial [14], antiviral [15], and antithrombotic [16] properties, among others. Despite all the environmental, economic, and nutritional advantages associated with the introduction of insects in human diet, there are some health risks that demand their careful assessment, such their antinutrient contents [17] and the possibility of causing adverse allergic reactions [18].

More than 2300 insect species of 18 orders are considered edible [19], but to date the European Union (EU) has only authorized the placing on the market of four species of insects that comply with the legislation [20] on novel foods for human consumption, namely, *Tenebrio molitor* larvae (yellow mealworm) [21,22], *Locusta migratoria* (migratory locust) [23], and *Acheta domesticus* (house cricket) [24], and *Alphitobius diaperinus* larvae (lesser mealworm) [25]. The establishment of legislation ensuring the safety of insects for human consumption as food and their availability as insect flours are two factors in favor of their general acceptability. Although the consumption of insects has been well correlated with some health benefits, being recognized by the traditional medicine for centuries, more scientific data are needed to support and increase consumers’ acceptance. This systematic review is aimed at performing an exhaustive bibliographic search of all research articles reporting sequenced bioactive peptides obtained from edible insects and the respective properties demonstrated by in silico, in vitro, and/or in vivo approaches. This report intends to evaluate the existing weigh-of-evidence regarding each specific claimed bioactive property, thus representing a valuable contribution to the divulgation of the scientific basis on the health benefits associated to the consumption of insects. As far as we know, this is the first systematic review compiling all information about sequenced bioactive peptides obtained from edible insects. These data will be very useful to identify gaps, serving as a starting point for other research works that are much needed to test and validate, in vivo, the proposed bioactivities.

## 2. Materials and Methods

### 2.1. Search Strategy

This review was developed according to the Preferred Reporting Items for Systematic Reviews and Meta-Analyses (PRISMA) methodology [26]. The protocol was registered in INPLASY (International Platform of Registered Systematic Review and Meta-analysis Protocols) with the registration number INPLASY202330075.

The publications were retrieved from the PubMed, Web of Science, and SCOPUS databases on 9 August 2022, and the keywords entered were

((((((insect) OR (larva*)) AND (hydrol*)) OR (digest*)) AND (peptide*)) AND (enzym*)) AND (bioact*) in PubMed.

((((((ALL = (insect)) OR ALL = (larva*)) AND ALL = (hydrol*)) OR ALL = (digest*)) AND ALL = (peptide)) AND ALL = (enzym*)) AND ALL = (bioact*) in Web of science.

insect OR larva* AND hydrol* OR digest* AND peptide AND enzym* AND bioact* in SCOPUS.

The search returned a total of 7559 publications (680 from PubMed; 1541 from Web of Science; 5338 from Scopus). The tool “Document type” available in the Web of Science interface was used to select the option “articles”, reducing its output to 1280 publications. The tools “Document type” and “language” available in the Scopus interface were used to select the options “articles” and “English”, respectively, reducing its output to 2543 publications. Therefore, a total of 4503 publications from the three databases were inserted into the EndNote library, whose tools were applied to automatically recognize and eliminate 364 duplicates (Figure 1).

### 2.2. Exclusion Criteria and Results

The title and abstract of each of the 4139 articles compiled in the EndNote library were independently revised by two authors of this publication. The duplicates not automatically recognized by the EndNote software, the review articles, and all original studies focusing on other species rather than insects were excluded. The full text of 370 articles was reviewed. From those, only the studies focusing on edible insects and reporting the amino acid sequence of potentially bioactive peptides (n = 36) were selected for careful examination. The studies describing the sequence of peptides obtained from insect sub-products (e.g., *Bombyx mori* cocoon) were excluded (n = 4). However, 4 research articles, not returned from the database search, were identified through cross-citation search, thus totalizing 36 studies that were included in this review (Figure 1). All selection steps were performed with the agreement of all authors according to the exclusion and inclusion criteria initially established.

### 2.3. Data Extraction

All the information considered relevant for this review, including insect species, sample, sample treatment, type of study, type of gastrointestinal (GI) digestion, analytical methods for peptide identification, peptide sequence, bioactive property, enzyme/cellular/organ/animal target, in vitro assays, in vitro outputs, in vivo assays, in vivo outputs, in silico assays (software/database), and bibliographic reference, was retrieved from each research article and collected in the Excel spreadsheet available in the Appendix A.

## 3. Results

This systematic review is organized into two major sections, the first one is regarding the data available for the species of edible insects with identified bioactive peptides, while the second section focuses on the relevant bioactive properties. The insect species were listed according to their family classification presented in Table 1.

### 3.1. Species of Edible Insects with Sequenced Bioactive Peptides

In this review, 36 published articles were selected and included. The collected data identified and characterized bioactive peptides resulting from the hydrolysis of 12 different edible insect species, namely, *A. diaperinus* larvae (lesser mealworm), *T. molitor* larvae (yellow mealworm), *Polyphylla adspersa* larvae (white grub larvae), *Gryllodes sigillatu* (tropical banded cricket), *Gryllus assimilis* (black cricket), *Schistocerca gregaria* (desert locust), *Apis mellifera* larvae and pupae (honeybee), *Oecophylla smaragdina* larvae and pupae (weaver ant), *Bombyx mori* pupae (silkworm), *Spodoptera littoralis* larvae (cotton leafworm), *Hermetia illucens* larvae (black soldier fly), and *Musca domestica* larvae and pupae (housefly) (Figure 2). The general characteristics of each species, nutritional composition, and respective bioactive peptides are described and discussed in the following sections. The number of bioactive peptides by insect species and the type of study (in vitro and/or in vivo) that were used for the identification of each bioactive property are summarized in Table 1 and Table 2, and graphically represented in Figure 3.

#### 3.1.1. *Gryllodes sigillatus*

The *G. sigillatus*, commonly known as the tropical banded cricket, belongs to the Orthoptera order and to the Gryllidae family (Table 1). They are native to Southeast Asia, though currently spread worldwide [54]. Crickets are edible insects that are very easy to breed, whose large-scale farming can be readily implemented. Among the different cricket species, the *G. sigillatus* is the smallest one, presenting important advantages, namely, high fertility and superior resistance to viruses and fungi [55]. Notably, it has a high protein content when compared to other edible insect species, including other cricket species. Nutritionally, the dehydrated *G. sigillatus* is composed of 70.0% of protein, 18.2% of fat, 3.7% of fiber, and 0.1% of carbohydrates, with an energy content of 452 kcal/mol [56].

So far, there are three studies reporting the identification of seven bioactive peptides obtained after the in vitro simulated GI digestion of *G. sigillatus* hydrolysates (Table 1 and Table 2) [11,13,47]. Hall, Reddivari, and Liceaga [47] found three peptides (#109 to #111, Table 2) capable of inhibiting the ACE as assessed by in silico analysis, suggesting an antihypertensive bioactivity. The other two studies were performed by the same research group, who identified four multifunctional peptides (#8 to #11, Table 2) in four different *G. sigillatus* samples (raw, boiled, baked, and protein extract). Firstly, the impact of the thermal processing (boiling and baking) of *G. sigillatus* on the formation of peptides with antioxidant (antiradical activity) and anti-inflammatory activities (inhibition of the lipoxygenase (LOX) and cyclooxygenase-2 (COX)) was investigated [13]. Posteriorly, their ability to inhibit the enzymatic activities associated with the development of the metabolic syndrome, namely, ACE (antihypertensive), α-glucosidase (antidiabetic), and lipase (antiobesity), was evaluated [11]. It was suggested that the four peptides possess multifunctional antioxidant/anti-inflammatory/antihypertensive/antidiabetic and antiobesity properties. Although all the peptides were obtained through in vitro gastrointestinal digestion and tested in vitro, they require further in vivo validation.

#### 3.1.2. *Gryllus assimilis*

*G. assimilis*, commonly known as black cricket, belongs to the Orthoptera order and to the Gryllidae family (Table 1). It is native to Jamaica although it has currently spread to the Caribbean islands, south of Texas and Mexico [54]. The dehydrated *G. assimilis* is composed of 65.5% of protein, 21.8% of lipids, and 8.6% of dietary fibers [57].

The only study identifying peptides with bioactive properties in *G. assimilis* used different combinations of commercial enzymes (flavoenzyme and neutrase; flavoenzyme and alcalase) to obtain two hydrolysates (Table 1 and Table 2) [29]. The authors identified 25 peptides with potential bioactive properties predicted in silico. Thirteen peptides have potential antidiabetic properties resulting from the inhibition of α-amylase (#19, #137, #138, #139), α-glucosidase (#140 to #143, #146), and DPP-IV (#144, #145, #147). Twelve peptides prevent the ACE and have potential antihypertensive properties (#19, #112 to #122). One peptide was predicted to possess antioxidant properties (#82, Table 2). One peptide (#175) was predicted to inhibit the 3-hydroxy-3-methylglutaryl coenzyme A (HMG-CoA) reductase possessing potential hypocholesterolemic properties, while another one (#19, Table 2) was classified as multifunctional with antidiabetic (impedes α-amylase) and antihypertensive properties (hinders ACE). Thus far, the *G. assimilis* was the only insect whose hydrolysis generated a peptide that was predicted to act as an inhibitor of the HMG-CoA reductase, thus suggesting a hypocholesterolemic property [29]. Although the inhibitory activities of the enzymes α-amylase, α-glucosidase, and ACE were assessed in vitro in the two hydrolysates, the authors did not evaluate the formation and/or stability of the identified peptides after GI digestion. Additionally, the in silico predicted individual bioactivities of peptides were not confirmed by in vitro analysis.

#### 3.1.3. *Schistocerca gregaria*

The *S. gregaria*, known as the desert locust, belongs to the Orthoptera order and to the Acrididae family (Table 1). It is believed that this species is native to the continent of America and has migrated to northeast Africa, where it is now very common. It can fly up to 150 km a day and increase in number up to 8000 times in 9 months, which are two characteristics that turn this locust species into a pest of economic importance in several world regions [54]. These locusts are very common in African and Arabian diets, being consumed fried, roasted, or boiled. The dried *S. gregaria* has a composition of 76.0% of protein, 13.0% of fat, 2.53% of fiber, and 1.7% of carbohydrates, with an energy content of 432 kcal/100 g [56]. Its protein content is very high compared to the values obtained for *G. sigillatus* and *T. molitor* under the same experimental conditions [56], suggesting that *S. gregaria* is a very good source of proteins.

There are two studies reporting the identification of four multifunctional peptides (#12 to #15) (Table 1 and Table 2), obtained from the simulated GI digestion of four hydrolysates of *S. gregaria*: raw, boiled, baked, and protein extract [11,13]. The antioxidant (free radical-scavenging activity, ion chelating activity, and reducing power), anti-inflammatory (LOX and COX inhibition) [13], antihypertensive (ACE inhibition), antiobesity (lipase inhibition), and antidiabetic (α-glucosidase inhibition) properties of the peptides were demonstrated by in vitro assays [11].

#### 3.1.4. *Alphitobius diaperinus*

The *A. diaperinus*, known as the lesser mealworm, is a species of beetle that belongs to the Coleoptera order and Tenebrionidae family (Table 1). It is native to Sub-Saharan Africa, but currently it is a cosmopolitan species. Lesser mealworms are considered a pest in poultry farms because they are reservoirs of avian pathogens. Additionally, during their development, these insects use the thermal isolation of the building of the poultry farms as pupation sites, thus causing important economic losses for farmers [54]. However, their larvae are edible and it is expected that, early in 2023, *A. diaperinus* will become the fourth species obtaining authorization for commercialization and consumption within the EU [58]. The nutritional content of the lesser mealworm powder is 58.4% of protein, 26.3% of fat, 7.46% of fiber, and an energy content of 494.8 kcal/100 g [59].

From this systematic literature search (Table 1 and Table 2), only a single study identified 25 potentially bioactive peptides in two materials (raw larvae flour—LF and protein extract—PE) of *A. diaperinus* larvae hydrolyzed with artichoke (*Cynara scolymus* L.) enzyme extract [27]. Twenty peptides were identified in the LF hydrolysate, from which 17 were predicted to possess antioxidant activity (#20 to #36, Table 2) and five peptides in the PE hydrolysate were expected to have different bioactivities, namely, one antioxidant (#37, Table 2), one antihypertensive (#83), and three multifunctional (antioxidant/antihypertensive; #1 to #3, Table 2). The antioxidant and the antihypertensive activities were determined by in vitro 2,2-diphenyl-1-picrylhydrazyl (DPPH) and angiotensin-converting enzyme (ACE) inhibitory assays for both hydrolysates, but not for the individual peptides [27]. Since the bioactive properties of the isolated peptides were only theoretically predicted and the in vitro GI digestion was poorly simulated (trypsin as the only digestive enzyme used), further in vitro/in vivo studies are required to validate the suggested bioactivities.

#### 3.1.5. *Tenebrio molitor*

The *T. molitor*, known as the yellow mealworm or the mealworm beetle, belongs to the Coleoptera order and Tenebrionidae family (Table 1). This insect is native to the Mediterranean region, though it has been spreading to other territories with temperate climates where it is considered a pest for stored goods (e.g., wheat and maize flour) [54]. Its larvae are edible, being one of the three insect species that complies with the regulation on novel foods [20]. The legislation allows its commercialization within the UE in the frozen, dried, and powder forms [21,22]. The nutritional composition of the *T. molitor* dried larvae is 52.4% of protein, 24.7% of fat, 1.97% of fiber, 2.20% of carbohydrates, and an energy content of 444 kcal/100 g.

To date, there are eight studies identifying 27 bioactive peptides that result from *T. molitor* hydrolysates (Table 1 and Table 2) [9,11,13,15,16,48,50,53]. The first study reported the identification of a tripeptide (#129, Table 2) with ACE inhibitory properties as assessed by in vitro analysis. The same study also demonstrated in vivo that a single oral administration to spontaneously hypertensive rats of the hydrolysate fraction containing the identified peptide can lead to the reduction of their systolic blood pressure [50]. A few years later, the same peptide (#129, Table 2) was identified and characterized in a hydrolysate fraction obtained after the simulated GI digestion of *T. molitor* larvae, where the authors demonstrated in vitro ACE inhibition in four (#125, #126, #127, #129, Table 2) peptides with potential antihypertensive bioactivity [48].

Peptide fractions obtained from *T. molitor* subjected to different heat treatments (boiling and baking) exhibited antioxidant, anti-inflammatory [13], and inhibitory activities against key enzymes relevant to the metabolic syndrome: ACE (antihypertensive), α-glucosidase (antidiabetic), and lipase (antiobesity) [11]. Four potential multifunctional peptides with antioxidant, anti-inflammatory, antidiabetic, antiobesity, and anti-inflammatory characteristics were also identified (#16 to #19, Table 2) [11,13]. In a different study, 13 antidiabetic (α-glucosidase or DPP-IV inhibition) peptides were detected (#148 to #160), but their inhibitory ability was not evaluated in vitro [9]. For the first time, peptides with anti-SARS-CoV-2 [15], hepatoprotective [53], and antithrombotic [16] bioactivities were identified in the hydrolysates from *T. molitor*. Two peptides (#172 and #173, Table 2) with anti-SARS-CoV-2 properties were identified and characterized in silico by inhibiting SARS-CoV-2 spike glycoprotein, main protease, and papain-like protease, thus requiring further in vitro and in vivo validation [15]. Two peptides (#170 and #171, Table 2) were tested in alpha mouse liver 12 (AML12) cells to evaluate their hepatoprotective properties against hydrogen peroxide (H_2_O_2_)-induced cytotoxicity. Although the authors were able to demonstrate the bioactivity of the peptides, they did not evaluate the effect of the GI digestion on their formation and/or stability [53], meaning that it is unknown whether these peptides can reach the lumen in an intact state. Another study reported the identification of two peptides (#168 and #169, Table 2) in a protein fraction possessing in vitro antithrombotic activity, which might be justified by the interaction of the peptides with thrombin exosite 1, as evaluated by in silico molecular docking [16].

#### 3.1.6. *Polyphylla adspersa*

The *P. adspersa*, known as the white grub, belongs to the Coleoptera order and the Scarabaeidae family (Table 1). It can be found in the territory of the extinguished Soviet Union, in the north of Turkey and the north of Iran [36]. Its larvae is a soil pest that eats roots, leaves, and young fruits. Its action usually culminates with the death of a broad range of fruit trees and crops, leading to severe economic impact damages. The larval stages of *P. adspersa* have a lifespan of 2–3 years [60] and their nutritional characteristics have not been studied so far. The determination of its protein content would be important to evaluate its potential viability as an alternative protein source.

In the only study devoted to *P. adspersa* as a source of bioactive peptides (Table 1 and Table 2), two bioactive peptides were extracted and purified (#80 and #81, Table 2) from the larvae hydrolysates and their antioxidant properties on adenocarcinomic human alveolar basal epithelial cells (A549) were tested [36]. The authors concluded that the peptides exerted their antioxidant activity through intracellular reactive oxygen species (ROS) scavenging and by inducing the activities of the superoxide dismutase (SOD), catalase (CAT), and glutathione peroxidase (GPx) in A549 cells. None of these peptides demonstrated a significant toxicity on A549 cells, human umbilical vein endothelial cells (HUVECs), or human red blood cells [36]. However, the claimed bioactive properties were not confirmed after GI digestion to assess the formation or the stability of the identified peptides from *P. adspersa*.

#### 3.1.7. *Apis mellifera*

The *A. mellifera*, commonly known as the honeybee, belongs to the Hymenoptera order and to the Apidae family (Table 1). Honeybees are widely reared worldwide to produce honey, beeswax, royal jelly, propolis, and bee venom. The honeybee brood, comprising larvae, pupae, and eggs, is a byproduct of farming that is consumed in several parts of the world due to its high nutritional value [61]. The dried *A. mellifera* larvae has a nutritional composition of 35.3% of protein, 14.5% of fat, 46.1% of carbohydrate, and an energy content of 456 kcal/100 g. The dried pupae contain 45.9% of protein, 16.0% of fat, and 34.3% of carbohydrate, with an energy content of 465 kcal/100 g [62].

The production of bioactive peptides resulting from the digestion of *A. mellifera* larvae and pupae proteins was demonstrated (Table 1 and Table 2) by two studies using in vitro and/or in silico approaches, respectively [37,38]. In the first report, one potential antihypertensive peptide (#84, Table 2) was isolated and identified after the hydrolysis of larva proteins through in vitro GI digestion [37]. In the second report, three novel antihypertensive peptides (#85, #86, #87, Table 2) from pupae hydrolysates were found and evaluated for their resistance to proteolysis through a simulated in silico GI digestion [38]. The antihypertensive properties of the four individual peptides, identified in the two studies, were evaluated through in vitro ACE inhibitory assays. Further studies are required to assess the bioavailability of peptides and confirm their in vivo antihypertensive properties.

#### 3.1.8. *Oecophylla smaragdina*

*O. smaragdina*, known as the Asian weaver ant, belongs to the Hymenoptera order and to the Formicidae family (Table 1). It is an arboreal ant species found in tropical regions of Asia and Australia. The mixture of *O. smaragdina* larvae and pupae is used in the traditional culinary of several Asian countries (e.g., Thailand and India) [35]. The dried *O. smaragdina* contains 55.3% of protein, 15.0% of fat, 19.9% of fiber, 7.30% of carbohydrates, and has an energy content of 385 kcal/100 g [63].

The only study focusing on *O. smaragdina* as a source of bioactive peptides identified three peptides (#79, #123, #124, Table 2), resulting from the in vitro GI digestion of a mix of larvae and pupae of weaver ants (Table 1 and Table 2) [35]. Two peptides (#123, #124, Table 2) exhibited antihypertensive properties, as demonstrated by in vitro and in silico tests based on their capability to inhibit the ACE. The other peptide (#79, Table 2) possessed antioxidant properties, which were in vitro evaluated through an 2,2′-azinobis-(3-ethylbenzothiazoline-6-sulfonic acid) (ABTS) radical scavenging assay [35].

#### 3.1.9. *Bombyx mori*

The *B. mori* is commonly known as the domesticated silkworm and belongs to the Lepidoptera order and to the Bombycidae family (Table 1). The cultivation of silkworms (sericulture) to produce silk fibers is a practice that originated in China many centuries ago. The most extensively produced silk is the one spun by the *B. mori*, which feeds on white mulberry leaves (*Morus alba*). This species became entirely dependent on humans and no longer occurs naturally in the wild [64]. Silk is obtained from the cocoon generated during the transformation of silkworm larvae to pupae phase (metamorphose), the silkworms being a subproduct produced in large quantities by the textile industry. In addition, *B. mori* pupae are frequently used in food, feed, and traditional medicine in Asian countries, being the only insect in the list of novel food resources published by the Ministry of Health of China [65]. The nutritional value of the dehydrated *B. mori* pupae was reported to present a content of 45–60% of protein, 20–35% of fat, 0.92–28.2% of carbohydrates, and 0.54–6.38% of fiber [66].

Several bioactive peptides (Table 1 and Table 2) were identified in *B. mori*. From the 36 selected reports, 15 focused on *B. mori* pupae [12,14,30,31,32,39,40,41,42,43,44,45,46,51,52], while one did not specifically target the pupae stage [46]. A total of 57 peptides with potential bioactive properties were identified in this species, although for 16 peptides it was not possible to predict their specific biological effect [14]. Twenty-one peptides (#88 to #108, Table 2) were predicted to have antihypertensive properties, from which 17 (#88 to #104, Table 2) were demonstrated to inhibit the activity of ACE [39,40,41,42,43,44,45,46]. Moreover, seven peptides (#161 to #167, Table 2) with in silico predicted antimicrobial activity [14] and six peptides (#38 to #43, Table 2) with in vitro antioxidant activities were identified in *B. mori* [30,31,32]. Two of them (#38, #39, Table 2) were able to reduce the formation of ROS by 40% in 2,2′-azobis (2-amidinopropane) dihydrochloride (AAPH)-induced human hepatoma (HepG2) cells, compared to untreated control HepG2 cells. However, this study lacked a simulated GI digestion assessment to verify their formation and resistance towards proteolytic enzymes [30]. Six peptides showed antidiabetic properties through in vitro α-glucosidase (#131 to #134, Table 2) [51] or dipeptidyl peptidase-IV (DPP-IV) (#135 and #136, Table 2) [52] inhibition tests. One peptide (#175, Table 2) demonstrated the ability to stimulate in ~248.4% the proliferation of 6-week-old imprinting control region (ICR) mice splenocytes induced by concanavalin A (Con A) and lipopolysaccharide (LPS), suggesting immunomodulatory properties [12].

The *B. mori* is the insect species with the highest number of identified bioactive peptides, and the only one with a potential immunomodulatory peptide. However, to date, there is no study demonstrating the bioavailability and biological effects of any of those peptides in vivo.

#### 3.1.10. *Spodoptera littoralis*

The *S. littoralis*, known as cotton leafworm, is a moth that belongs to the Lepidoptera order and to the Noctuidae family (Table 1). This species is native to Africa, but it can also be found in the subtropical regions of Europe, America, and Africa. *S. littoralis* is polyphagous and considered a plague of several important crops, thus being able to provoke major damages to local economies. Its lifecycle comprises egg, larvae, pupa, and adult phases. The nutrient composition of the dried *S. littoralis* larvae includes 51.2% of protein, 33.1% of fat, 5.2% of carbohydrates, and 10.7% of fiber [67].

There are two studies from the same research group that evaluated the bioactivity of peptides obtained from a hydrolysate of *S. littoralis* larvae (Table 1 and Table 2) [8,49]. In the first study, one potentially antihypertensive peptide (#128, Table 2) was identified after the simulated GI digestion of *S. littoralis* larvae, whose ACE inhibitory activity was demonstrated in vitro [49]. In a posterior study, the same group performed additional in vitro tests to corroborate the antihypertensive characteristics of that tripeptide (#128, Table 2) and its dipeptide (#130, Table 2) fragment, which validated its bioactivity in vivo. The ACE inhibitory activity of the dipeptide was detected in organ bath experiments using isolated rat aorta, and the antihypertensive bioactivity of both peptides was in vivo validated through its oral administration to spontaneously hypertensive rats. The results showed that the ingestion of both peptides led to a significant decrease in blood pressure [8]. This is the only in vivo study demonstrating the antihypertensive activity of individual peptides obtained from the GI digestion of insect proteins.

#### 3.1.11. *Hermetia illucens*

The *H. illucens*, commonly known as the black soldier fly, belongs to the Diptera order and to the Stratiomyidae family (Table 1). Although native to the Neotropical region, it became a cosmopolitan species found in the wild in multiple countries with temperate climates [54]. Its lifecycle consists of four stages: egg, larvae, pupae, and adult. It has been suggested that the *H. illucens* adults are nonfeeding and do not possess a functional gut, but a recent study demonstrated that their longevity increases when they are fed [68]. Their larvae are saprophytic as they primarily feed on organic wastes, converting these materials into fat, protein, chitin, and vitamins that are stored in their body to support their metabolism during the pupal and adult stages [69]. Their waste-processing ability is a highly relevant characteristic, which is currently being explored for industrial-scale applications [70]. The nutritional composition of *H. illucens* varies according to their development stage, with dehydrated prepupae containing 43.7% of protein, 31.8% of fat, 10.1% of fiber, 12.3% of carbohydrates, and 575 kcal/100 g of energy [69].

Two reports identified 33 peptides with antioxidant properties obtained from hydrolysates of *H. illucens* larvae (Table 1 and Table 2) [10,33]. One of the studies evaluated the antioxidant potential of a pool of 17 peptides (#44 to #60, Table 2) through in vitro methodologies (2,2-diphenyl-1-picrylhydrazyl (DPPH), hydroxyl radicals, superoxide, and ABTS radical scavenging activities and reducing power) [10]. The other study predicted the bioactivity of 16 peptides (#61 to #76, Table 2) obtained from the hydrolysate of *H. illucens* larvae fed with food wastes [33] by in silico tools. None of the studies evaluated the effect of the GI digestion in the formation or in the stability of the identified peptides or assessed their individual bioactive properties by in vitro assays.

#### 3.1.12. *Musca domestica*

The *M. domestica*, commonly known as the housefly, belongs to the Diptera order and to the Muscidae family (Table 1). Although native to Central Asia, houseflies have spread to any place inhabited by humans. It is a cosmopolitan saprophagous species with a high reproductive capability, attracted by humans and animals. In some places, they are considered a public health problem due to their capability to carry pathogens (e.g., *Salmonella enteritidis*, *Escherichia coli*, Campylobacter spp.) infecting humans, farm animals, and pets [54]. The housefly larvae and pupae have been used in Chinese traditional medicine for hundreds of years to treat several health conditions (gastrointestinal disease, wound healing, damp-heat diarrhea, vomiting, etc.) [34]. Dried housefly larvae and pupae are composed of 60.4% and 76.2% of protein, 14.1% and 14.4% of fat, and 8.6% and 15.7% of fiber, respectively [71].

Seven bioactive peptides in *M. domestica* pupae and larvae were identified in two reports, respectively (Table 1 and Table 2) [28,34]. One of the studies identified and characterized two peptides (#77 and #78, Table 2) with antioxidant properties in housefly pupae hydrolysate. The in vitro studies demonstrated that both peptides are resistant to simulated GI digestion and exhibited potent antioxidant activity and neuroprotective capacity against H_2_O_2_-induced oxidative stress damage in rat pheochromocytoma (PC12) cells. One of the peptides was capable to effectively protect PC12 cells from oxidative damage induced by H_2_O_2_ by decreasing malonaldehyde (MDA) and intracellular ROS, increasing the activity of intracellular SOD and recovering cellular mitochondrial membrane potential (MMP) [34]. The other study used an in silico approach to simulate the GI digestion of eight major housefly larvae proteins. The authors identified five dipeptides (#4, #5, #6, #7, #130, Table 2) with predicted multifunctional antihypertensive (inhibit ACE) and antidiabetic (inhibit DPP-IV) properties [28]. Peptides #4 to #7 (Table 2) require further in vitro validation, but the antihypertensive properties of peptide VF (#130, Table 2) were evaluated in vitro and in vivo in a previously reported study focusing on *S. littoralis* [8].

### 3.2. Bioactive Properties of the Identified Peptides

From the 36 articles included in this systematic review (Figure 1), 211 potentially bioactive peptides were identified in 12 insect species (Appendix A). From them, 19 peptides were predicted, by in silico tools, to possess some type of bioactivity, but without assessing the specific property [14,27]. Fifteen peptides (#8 to #19, #128 to #130, Table 2) were identified and/or characterized in more than one article. All these peptides were described in different articles focusing on the same insect species, except for peptides #19 and #130 (Table 2) that were identified in more than one insect species. The GDDAPR peptide (#19, Table 2) was identified and characterized as having multifunctional properties: antioxidant/anti-inflammatory [13] and antihypertensive/antidiabetic/antiobesity in *T. molitor* hydrolysates [11]; and antihypertensive/antidiabetic in *G. assimilis* hydrolysates [29]. Peptide VF (#130, Table 2) was recently identified as antidiabetic/antihypertensive in *M. domestica* hydrolysates [28], with its antihypertensive property being previously validated in vivo in *S. littoralis* larvae hydrolysates [8].

Overall, there is a total of 175 peptides whose bioactive properties were characterized by in silico and/or in vitro and/or in vivo approaches (Table 2). Peptides with antioxidant, antihypertensive, antidiabetic, antiobesity, anti-inflammatory, hypocholesterolemic, antimicrobial, anti-SARS-CoV-2, antithrombotic, and immunomodulatory properties were identified (Figure 4). Twenty peptides are multifunctional with antioxidant/antihypertensive (#1 to #7, Table 2), antioxidant/anti-inflammatory/antihypertensive/antidiabetic/antiobesity (#8 to #19, Table 2), and antidiabetic/antihypertensive (#130, Table 2) properties. A brief description of each biological activity associated with the identified peptides and respective health benefits related to their consumption is provided in the following subsections.

#### 3.2.1. Antioxidant Peptides

Although all living organisms possess a system of enzymatic and nonenzymatic antioxidant agents to prevent the oxidative damage, the consumption of dietary sources of antioxidants is very important to provide additional protection to balance the oxidative status [72]. This systematic review clearly shows that insects are a good source of antioxidant peptides. Eighty peptides with antioxidant properties (#1, #2, #3, #8 to #82, #170, and #171, Table 2) were identified in 10 out of 12 insect species referred to in this systematic review. The species without potential antioxidant peptides are *A. mellifera* and *S. littoralis*. The in vitro evaluation of the oxidant activities of the peptides was performed, in most cases, by assessing the DPPH and ABTS scavenging ability of the hydrolysates. Only 25 potential antioxidant peptides (#8 to #19, #38 to #43, #77 to #81, #170, #171, Table 2) were individually characterized in vitro, and from these, eight (#38 #39, #77, #78, #80, #81, #170, #171, Table 2) were further tested in different cell lines, as reported by four relevant studies.

The peptides NDVLFF and SWFVTPF (#38 and #39, Table 2) were identified in *B. mori* hydrolysates and their antioxidant properties were demonstrated through in vitro antioxidant assays and an in situ assay for ROS reduction using hepatic HepG2 cells [30]. The peptides DFTPVCTTELGR and ARFEELCSDLFR (#77 and #78, Table 2) were identified and characterized in a hydrolysate of *M. domestica* pupae and proved to exert strong ABTS cation radical scavenging ability. The peptide ARFEELCSDLFR (#78, Table 2) was able to effectively protect PC12 cells from oxidative damage induced by H_2_O_2_ by decreasing the intracellular ROS and MDA, recovering cellular MMP, and increasing the activity of intracellular SOD [34]. The peptides YPQSLRWRAK and LPLFFYDVRP (#80 and #81, Table 2), isolated from *P. adspersa* larvae hydrolysate, were able to protect adenocarcinoma human alveolar basal epithelial A549 cells against free radical damages. None of the peptides demonstrated a significant toxicity effect on A549 cells, HUVECs, and human red blood cells [36]. The peptides AKKHKE and LE (#170 and #171, Table 2), isolated from protein hydrolysates of *T. molitor* larvae, showed a high protective effect against H_2_O_2_-induced cytotoxicity in AML12 mouse hepatocyte cells [53].

#### 3.2.2. Antihypertensive Peptides

Hypertension is a chronic health condition disorder in blood pressure that significantly increases the risk of heart attack, stroke, and kidney damage, among other health problems. According to the data provided by WHO, the number of adults with hypertension increased from 594 million in 1975 to 1.13 billion in 2015 (https://www.who.int/news-room/fact-sheets/detail/hypertension, accessed on 30 October 2022). The ACE (EC 3.4.15.1) inhibitors are the most indicated drugs for first-line antihypertensive therapy [73,74]. The ACE is an enzyme associated with the rennin–angiotensin system that hydrolyzes angiotensin I to the octapeptide angiotensin II, resulting in arterial constriction and, consequently, in blood pressure elevation. Although its use is not devoid of some controversy, it is well established that ACE inhibition effectively lowers the systolic and diastolic blood pressure in both hypertensive and normotensive subjects [75].

The antihypertension capability is the most studied bioactive property in food-derived peptides [76] and the edible insects are not an exception. From the 36 studies selected for this review, 20 identified bioactive peptides with antihypertensive potential. A total of 67 peptides (#1 to #19, #83 to #130, Table 2) with potential antihypertensive bioactivity, including 20 multifunctional (#1 to #19, #130, Table 2), were identified in 10 out of the 12 insect species under study. From these, so far, any peptide with potential antihypertensive properties was identified in two species (*H. illucens* and *P. adspersa*). In general, the antihypertensive properties of the identified peptides were assessed by their ability to inhibit the ACE based on in silico and/or in vitro approaches. Forty-one (#8 to #19, #84 to #104, #123 to #130, Table 2) of the 67 peptides predicted to possess antihypertensive properties were individually characterized and their half maximal inhibitory concentration (IC_50_) was calculated. The antihypertensive activity of three peptides, namely, YAN (#129, Table 2), AVF (#128, Table 2), and VF (#130, Table 2), was validated in vivo by their administration (isolated or contained in a protein fraction) to spontaneously hypertensive rats [8,50].

#### 3.2.3. Antidiabetic Peptides

According to the WHO, more than 422 million people live with diabetes. It is an endocrine disorder characterized by hyperglycemia, resulting from impaired insulin secretion (type 1 diabetes) or altered insulin sensitivity (type 2 diabetes). More than 95% of people with diabetes have type 2 diabetes (https://www.who.int/news-room/fact-sheets/detail/diabetes, accessed on 13 October 2022). The contemporary therapeutic approach to diabetes is to decrease postprandial hyperglycemia (plasma glucose concentrations after eating), which can be accomplished through the inhibition of DPP-IV, α-glucosidase, and/or α-amylase enzymes [77]. DPP-IV (E.C. 3.4.14.5) is an exopeptidase with two natural substrates that include incretin hormones, glucagon-like peptide 1 (GLP-1), and glucose-dependent insulinotropic polypeptide (GIP), whose function is to enhance the glucose-induced insulin secretion during a meal [78]. The DPP-IV inactivates both hormones, restricting their half-life. Therefore, the inhibition of the DPP-IV activity promotes glucose-dependent insulin secretion and attenuates postprandial hyperglycemia [79]. The pancreatic α-amylase (E.C. 3.2.1.1) and α-glucosidase (EC 3.2.1.20) are digestive enzymes that cleave dietary carbohydrates (e.g., starch or table sugar) into simple monosaccharides, allowing them to be absorbed and enter the bloodstream. The inhibition of these enzymes can suppress carbohydrate digestion, delay glucose uptake, and, consequently, attenuate postprandial hyperglycemia [80].

A total of 47 peptides (#4 to #19″ #13′ to #160, Table 2) with antidiabetic properties were identified in 6 out of the 12 insect species focused on this systematic review: *B. mori*, *G. sigillatus*, *G. assimilis*, *M. domestica*, *S. gregaria*, and *T. molitor*. Twenty-seven peptides (#8 to #19, #131 to #134, #140 to #143, #146, #155 to #160, Table 2) were proposed as potential α-glucosidase inhibitors, and 16 (#8 to #19, #31 to #34) (Table 2) of them were in vitro characterized and their IC_50_ was calculated. Seventeen peptides (#4 to #7, #130, #135, #136, #144, #145, #147 to #154, Table 2) were proposed as DPP-IV inhibitors, but only two (#135 and #136, Table 2) were individually in vitro characterized and their IC_50_ calculated. Four potential α-amylase inhibitors (#19 and #137 to #139, Table 2) were identified, but their individual inhibitory activity was not evaluated in vitro. Peptide #19 (Table 2) is suggested to inhibit both α-glucosidase and α-amylase. There is no record of any study evaluating the antidiabetic activity of any of those peptides in cells or validating their activity in vivo.

#### 3.2.4. Other Bioactive Peptides

Although the antioxidant, antihypertensive, and antidiabetic are the most relevant properties among the identified peptides in edible insect species, some studies found other bioactivities, namely, antiobesity, anti-inflammatory, hypocholesterolemic, antimicrobial, anti-SARS-CoV-2, antithrombotic, and immunomodulatory.

Twelve peptides (#8 to #19) with multifunctional activities, including antiobesity and anti-inflammatory properties, were identified in different samples (raw, boiled, baked, protein) of *G. sigillatus*, *S. gregaria*, and *T. molitor* [11,13]. All the identified peptides firstly showed antioxidant potential and the ability to inhibit the enzymes LOX and COX related to inflammatory processes [13]. Inflammation is a natural protective response of the body triggered by a potentially harmful stimulus. The overproduction of mediators of arachidonic acid (AA) cascade, particularly those of LOX and COX pathways, is related to several inflammatory diseases and, therefore, the LOX and COX inhibitors are considered good inflammatory agents [81]. Afterwards, the same peptides were demonstrated to act as inhibitors of the three enzymes associated with the development of the metabolic syndrome: ACE, α-glucosidase, and lipase [11]. Obesity is a health issue caused by excessive fat absorption and accumulation. The pancreatic lipase is a metabolic enzyme that catalyzes the hydrolysis of triacylglycerols into small molecules of glycerol and fatty acids, which can be absorbed by the intestine and enter the bloodstream. The pancreatic lipase inhibitors attenuate the lipase activity, enabling lowering of the levels of lipids absorbed in the digestive tract and, consequently, reduction of the accumulation of adipose tissue [82]. The LOX, COX, and lipase inhibitory potential of the 12 identified peptides was characterized in vitro and the respective IC_50_ values were reported [11,13].

Two peptides (#168 and #169, Table 2) with potential antithrombotic activity were isolated from *T. molitor* larvae. Thrombosis occurs when a blood clot is formed inside an artery or vein, limiting the blood circulation. The acute arterial and venous thromboses are the most common cause of death in developed countries [83]. The blood coagulation is a complex physiological process controlled by a cascade of proteolytic reactions. Thrombin is one of the enzymes involved in the coagulation cascade and a main target for antithrombotic drugs [84]. The antithrombotic potential of both peptides was assessed in vitro through the evaluation of its potential to inhibit the thrombin exosite 1 [16].

A recent in silico study identified two peptides (#172 and #173, Table 2) as potential inhibitors of the spike glycoprotein, the main protease and papain-like protease of the SARS-CoV-2 [15]. Seven peptides (#161 to #167, Table 2) with antimicrobial properties were also identified through in silico tools in *B. mori* pupae [14]. One peptide (#174, Table 2) obtained from *G. assimilis* hydrolysate was identified by in silico tools as a potential HMG-CoA reductase inhibitor; however, its hypocholesterolemic potential requires further in vitro tests [29]. Another peptide (#175, Table 2) obtained from ultramicro-pretreated silkworm pupae protein and resistant to simulated GI digestion was proposed to have immunomodulatory properties. The in vitro experiments showed that the peptide was able to promote splenocyte proliferation induced by Con A or LPS [12]. Splenocytes consist of a variety of white blood cells in the spleen (e.g., macrophages, dendritic cells, T- and B-lymphocytes) that have different immune functions and are frequently used to evaluate immune responses.

## 4. Conclusions and Future Perspectives

The use of insects as a sustainable protein source has been widely discussed in recent years. The food industry tries to overcome the question of the repugnance caused by the insects’ physiognomy through its conversion into flours. Another possible approach to change the perception of consumers is to invest in the study and divulgation of the health benefits associated with the consumption of insects. Aligned with this idea, this systematic review gathers all the species in which peptides with bioactive properties have been identified and/or characterized, highlighting their potential role in the prevention of important health conditions. This systematic review identified 36 studies focused on the identification and/or characterization of bioactive peptides obtained from the hydrolysis of proteins from edible insects. In those reports, 12 edible insect species were targeted, from which the most studied are *B. mori* and *T. molitor*, two species regulated in China and EU, respectively, as novel foods. Several peptides with antioxidant, antihypertensive, antidiabetic, antiobesity, anti-inflammatory, hypocholesterolemic, antimicrobial, anti-SARS-CoV-2, antithrombotic, and immunomodulatory properties were reported. A total of 211 potentially bioactive peptides were identified, but only 61 were characterized individually through in vitro methodologies to evaluate their predicted bioactive properties. An important step in the identification of potential bioactive peptides is the simulation of GI digestion to evaluate their formation or stability. From the 211 identified peptides, 147 were obtained or submitted to a simulated GI digestion. In vitro tests and simulated GI digestion are key issues in the prediction of potentially bioactive peptides and in the evaluation of their resistance to proteolytic enzymes. However, these results are only predictive, since to be effectively bioactive, peptides need to be absorbed in the GI tract, cross the intestinal barrier, enter the bloodstream, and come into contact with molecular targets/receptors, a process that can only be evaluated in vivo. From the identified peptides, only three (YAN, AVF, and VF) were tested in vivo, all demonstrating effective antihypertensive properties in spontaneously hypertensive rats. This systematic review summarizes, for the first time, all the scientific evidence about the potential health benefits associated with the ingestion of insect proteins. The collected information suggests that the consumption of insects can play a key role in the prevention of some important pathologies.

## Figures and Tables

**Figure 1 foods-12-02026-f001:**
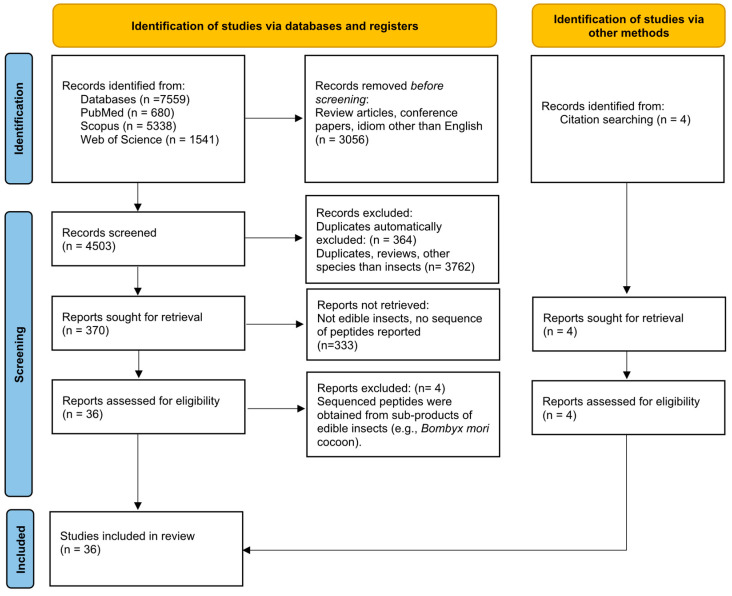
PRISMA 2020 flow diagram [26] of this systematic review.

**Figure 2 foods-12-02026-f002:**
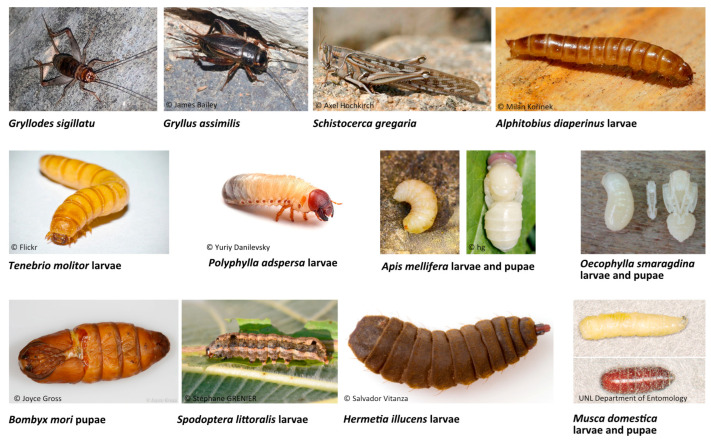
Photographs of the 12 insect species with bioactive peptides sequenced and characterized.

**Figure 3 foods-12-02026-f003:**
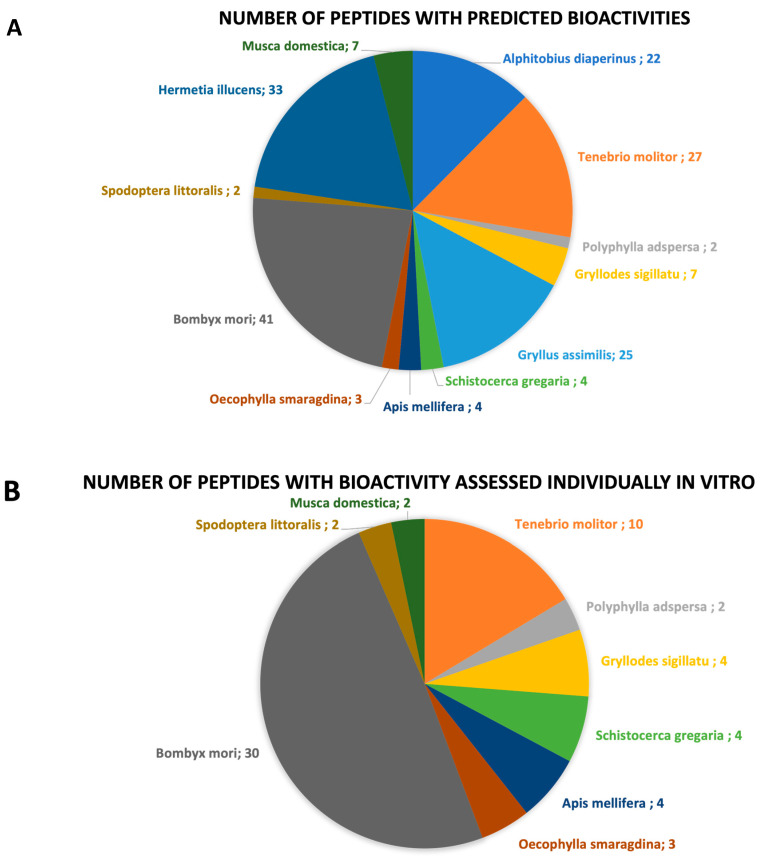
Chart representation of the (**A**) number of bioactive peptides with predicted bioactivities by insect species; (**B**) number of peptides whose bioactive properties were assessed individually through in vitro studies by insect species.

**Figure 4 foods-12-02026-f004:**
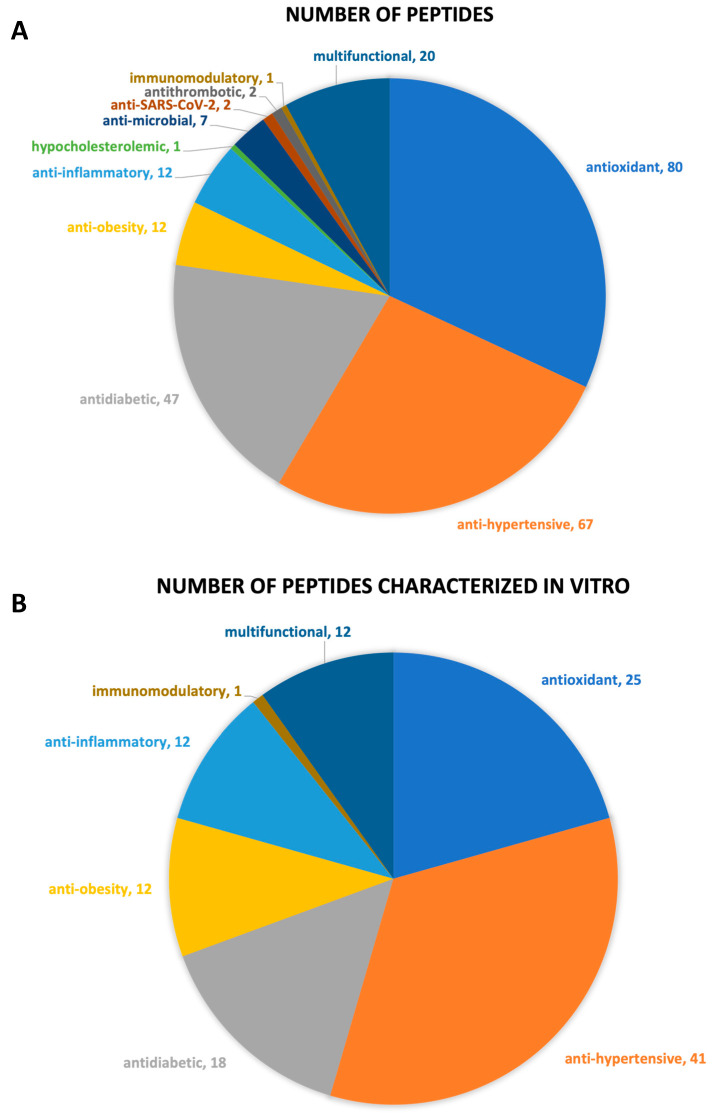
Chart representation of the (**A**) number of peptides by bioactive property; (**B**) number of peptides whose bioactive property was assessed through in vitro studies.

**Table 1 foods-12-02026-t001:** Summary of the number of peptides with predicted bioactivities identified in edible insect species.

Order	Family	Species	Number of Studies	Number of Peptides with Predicted Bioactivities	Number of Peptides with Bioactivity Assessed In Vitro	Number of Peptides with Bioactivity Assessed In Vivo
Orthoptera	Gryllidae	*Gryllodes sigillatu*	3	7	4	_
		*Gryllus assimilis*	1	25 ^a^	_	_
	Acrididae	*Schistocerca gregaria*	2	4	4	_
Coleoptera	Tenebrionidae	*Alphitobius diaperinus*	1	22	_	_
		*Tenebrio molitor*	8	27 ^a^	10	1
	Scarabaeidae	*Polyphylla adspersa*	1	2	2	_
Hymenoptera	Apidae	*Apis mellifera*	2	4	4	_
	Formicidae	*Oecophylla smaragdina*	1	3	3	_
Lepidoptera	Bombycidae	*Bombyx mori*	15	41	30	_
	Noctuidae	*Spodoptera littoralis*	2	2	2	2
Diptera	Stratiomyidae	*Hermetia illucens*	2	33	_	_
	Muscidae	*Musca domestica*	2	7 ^b^	2	_
Total			40 ^c^	177	61	3

^a,b^ one peptide was identified in both insect species (177-2 repeated peptides = 175 different peptides). ^c^ two studies are common to 3 insect species (40-4 repeated studies = 36 different studies).

**Table 2 foods-12-02026-t002:** Summary of the main characteristics of the bioactive peptides identified in insect hydrolysates, namely, peptide sequence, bioactive property, and insect species where they can be found.

	Peptide Sequence	Bioactive Property	Species	Bioactivity Assessed Individually In Vitro	Simulated GI Digestion	Reference
#1	APVAVAHAAVPA	antioxidant/antihypertensive	*A. diaperinus*	no/no	yes	[27]
#2	ASVVEKLGDY	antioxidant/antihypertensive	*A. diaperinus*	no/no	yes	[27]
#3	GLIGAPIAAPIAA	antioxidant/antihypertensive	*A. diaperinus*	no/no	yes	[27]
#4	AF	antidiabetic/antihypertensive	*M. domestica*	no/no	yes	[28]
#5	GW	antidiabetic/antihypertensive	*M. domestica*	no/no	yes	[28]
#6	GY	antidiabetic/antihypertensive	*M. domestica*	no/no	yes	[28]
#7	PH	antidiabetic/antihypertensive	*M. domestica*	no/no	yes	[28]
#8	IIAPPER	antioxidant/anti-inflammatory	*G. sigillatus*	yes/yes	yes	[13]
		antihypertensive/antidiabetic/antiobesity	*G. sigillatus*	yes/yes/yes	yes	[11]
#9	LAPSTIK	antioxidant/anti-inflammatory	*G. sigillatus*	yes/yes	yes	[13]
		antihypertensive/antidiabetic/antiobesity	*G. sigillatus*	yes/yes/yes	yes	[11]
#10	VAPEEHPV	antioxidant/anti-inflammatory	*G. sigillatus*	yes/yes	yes	[13]
		antihypertensive/antidiabetic/antiobesity	*G. sigillatus*	yes/yes/yes	yes	[11]
#11	KVEGDLK	antioxidant/anti-inflammatory	*G. sigillatus*	yes/yes	yes	[13]
		antihypertensive/antidiabetic/antiobesity	*G. sigillatus*	yes/yes/yes	yes	[11]
#12	GKDAVIV	antioxidant/anti-inflammatory	*S. gregaria*	yes/yes	yes	[13]
		antihypertensive/antidiabetic/antiobesity	*S. gregaria*	yes/yes/yes	yes	[11]
#13	AIGVGAIER	antioxidant/anti-inflammatory	*S. gregaria*	yes/yes	yes	[13]
		antihypertensive/antidiabetic/antiobesity	*S. gregaria*	yes/yes/yes	yes	[11]
#14	FDPFPK	antioxidant/anti-inflammatory	*S. gregaria*	yes/yes	yes	[13]
		antihypertensive/antidiabetic/antiobesity	*S. gregaria*	yes/yes/yes	yes	[11]
#15	YETGNGIK	antioxidant/anti-inflammatory	*S. gregaria*	yes/yes	yes	[13]
		antihypertensive/antidiabetic/antiobesity	*S. gregaria*	yes/yes/yes	yes	[11]
#16	NYVADGLG	antioxidant/anti-inflammatory	*T. molitor*	yes/yes	yes	[13]
		antihypertensive/antidiabetic/antiobesity	*T. molitor*	yes/yes/yes	yes	[11]
#17	AAAPVAVAK	antioxidant/anti-inflammatory	*T. molitor*	yes/yes	yes	[13]
		antihypertensive/antidiabetic/antiobesity	*T. molitor*	yes/yes/yes	yes	[11]
#18	YDDGSYKPH	antioxidant/anti-inflammatory	*T. molitor*	yes/yes	yes	[13]
		antihypertensive/antidiabetic/antiobesity	*T. molitor*	yes/yes/yes	yes	[11]
#19	AGDDAPR	antioxidant/anti-inflammatory	*T. molitor*	yes/yes	yes	[13]
		antihypertensive/antidiabetic/antiobesity	*T. molitor*	yes/yes/yes	yes	[11]
		antihypertensive/antidiabetic	*G. assimilis*	no/no	no	[29]
#20	GLIGAPIAAPI	antioxidant	*A. diaperinus*	no	yes	[27]
#21	AYVGPDGVTY	antioxidant	*A. diaperinus*	no	yes	[27]
#22	AESEVAALNR	antioxidant	*A. diaperinus*	no	yes	[27]
#23	GLIGAPIAAPIA	antioxidant	*A. diaperinus*	no	yes	[27]
#24	VDAAVLEKLE	antioxidant	*A. diaperinus*	no	yes	[27]
#25	ASVVEKLGDYL	antioxidant	*A. diaperinus*	no	yes	[27]
#26	VDAAVLEKLEA	antioxidant	*A. diaperinus*	no	yes	[27]
#27	AGFAGDDAPRAVF	antioxidant	*A. diaperinus*	no	yes	[27]
#28	GLIGAPIAAPIAAPL	antioxidant	*A. diaperinus*	no	yes	[27]
#29	ASLEAEAKGKAEAL	antioxidant	*A. diaperinus*	no	yes	[27]
#30	AIANAAEKKQKAF	antioxidant	*A. diaperinus*	no	yes	[27]
#31	FSLPHAILRLDL	antioxidant	*A. diaperinus*	no	yes	[27]
#32	YALPHAILRIDL	antioxidant	*A. diaperinus*	no	yes	[27]
#33	VDAAVLEKLEAGF	antioxidant	*A. diaperinus*	no	yes	[27]
#34	GLIGAPIAAPIAAPLA	antioxidant	*A. diaperinus*	no	yes	[27]
#35	PADTPEVAAAKVAHA	antioxidant	*A. diaperinus*	no	yes	[27]
#36	LKVDDLAAELDASQ	antioxidant	*A. diaperinus*	no	yes	[27]
#37	VAYSPAAVVSH	antioxidant	*A. diaperinus*	no	yes	[27]
#38	NDVLFF	antioxidant	*B. mori*	yes	no	[30]
#39	SWFVTPF	antioxidant	*B. mori*	yes	no	[30]
#40	FKGPACA	antioxidant	*B. mori*	yes	yes	[31]
#41	SVLGTGC	antioxidant	*B. mori*	yes	yes	[31]
#42	AAEYPA	antioxidant	*B. mori*	yes	no	[32]
#43	AKPGVY	antioxidant	*B. mori*	yes	no	[32]
#44	VEEPPKEE	antioxidant	*H. illucens*	no	no	[10]
#45	EEKNPKATE	antioxidant	*H. illucens*	no	no	[10]
#46	PTTAPSATIN	antioxidant	*H. illucens*	no	no	[10]
#47	VEEPPKEEKNPK	antioxidant	*H. illucens*	no	no	[10]
#48	MAAGTNLLDTK	antioxidant	*H. illucens*	no	no	[10]
#49	ETKNDEASVEQIK	antioxidant	*H. illucens*	no	no	[10]
#50	RPEELGPNK	antioxidant	*H. illucens*	no	no	[10]
#51	FPGGETEALRR	antioxidant	*H. illucens*	no	no	[10]
#52	AGGGGGGGGGGGKNL	antioxidant	*H. illucens*	no	no	[10]
#53	IHKAGGGGGGGGGGGK	antioxidant	*H. illucens*	no	no	[10]
#54	HPERPIPEH	antioxidant	*H. illucens*	no	no	[10]
#55	DQAKAFLEKDNK	antioxidant	*H. illucens*	no	no	[10]
#56	NWDLKEVGGGALP	antioxidant	*H. illucens*	no	no	[10]
#57	SATTAIYMNALL	antioxidant	*H. illucens*	no	no	[10]
#58	KDNEEAEAKPT	antioxidant	*H. illucens*	no	no	[10]
#59	SLGGEMKQTAK	antioxidant	*H. illucens*	no	no	[10]
#60	LTSGSANATGSR	antioxidant	*H. illucens*	no	no	[10]
#61	GYGFGGGAGCLSMDTGAHLNR	antioxidant	*H. illucens*	no	no	[33]
#62	AGLQFPVGR	antioxidant	*H. illucens*	no	no	[33]
#63	HFQAPSHIR	antioxidant	*H. illucens*	no	no	[33]
#64	VGIKAPGIIPR	antioxidant	*H. illucens*	no	no	[33]
#65	GFIGPGVDVPAPDMGTGER	antioxidant	*H. illucens*	no	no	[33]
#66	SQINFPIGGPTER	antioxidant	*H. illucens*	no	no	[33]
#67	AVDSLVPIGR	antioxidant	*H. illucens*	no	no	[33]
#68	VVPSANRAMVGIVAGGGRIDKPILK	antioxidant	*H. illucens*	no	no	[33]
#69	GFKDQIQDVFK	antioxidant	*H. illucens*	no	no	[33]
#70	TQLEPPISTPHCAR	antioxidant	*H. illucens*	no	no	[33]
#71	TIRYPDPLIK	antioxidant	*H. illucens*	no	no	[33]
#72	SKIPFNVTPGSEQIR	antioxidant	*H. illucens*	no	no	[33]
#73	RIPFSHDDR	antioxidant	*H. illucens*	no	no	[33]
#74	VLVDGPLTGVPR	antioxidant	*H. illucens*	no	no	[33]
#75	GVEEDWLSAR	antioxidant	*H. illucens*	no	no	[33]
#76	IGGIGTVPVGR	antioxidant	*H. illucens*	no	no	[33]
#77	DFTPVCTTELGR	antioxidant	*M. domestica*	yes	yes	[34]
#78	ARFEELCSDLFR	antioxidant	*M. domestica*	yes	yes	[34]
#79	CTKKHKPNC	antioxidant	*O. smaragdina*	yes	yes	[35]
#80	YPQSLRWRAK	antioxidant	*P. adspersa*	yes	no	[36]
#81	LPLFFYDVRP	antioxidant	*P. adspersa*	yes	no	[36]
#82	WDDMEK	antioxidant	*G. assimilis*	no	no	[29]
#83	LEKDNALDRAAM	antihypertensive	*A. diaperinus*	no	yes	[27]
#84	LLKPY	antihypertensive	*A. mellifera*	yes	yes	[37]
#85	AVFPSIVGR	antihypertensive	*A. mellifera*	yes	yes	[38]
#86	PGKVHIT	antihypertensive	*A. mellifera*	yes	yes	[35]
#87	PPVLVFV	antihypertensive	*A. mellifera*	yes	yes	[35]
#88	ASL	antihypertensive	*B. mori*	yes	yes	[39]
#89	RYL	antihypertensive	*B. mori*	yes	yes	[40]
#90	GAMVVH	antihypertensive	*B. mori*	yes	yes	[41]
#91	KHV	antihypertensive	*B. mori*	yes	yes	[42] *
#92	APPPKK	antihypertensive	*B. mori*	yes	no	[43] *
#93	GNPWM	antihypertensive	*B. mori*	yes	yes	[44] *
#94	IF	antihypertensive	*B. mori*	yes	yes	[45]
#95	GD	antihypertensive	*B. mori*	yes	yes	[45]
#96	DA	antihypertensive	*B. mori*	yes	yes	[45]
#97	TE	antihypertensive	*B. mori*	yes	yes	[45]
#98	TA	antihypertensive	*B. mori*	yes	yes	[45]
#99	ES	antihypertensive	*B. mori*	yes	yes	[45]
#100	SS	antihypertensive	*B. mori*	yes	yes	[45]
#101	ST	antihypertensive	*B. mori*	yes	yes	[45]
#102	SD	antihypertensive	*B. mori*	yes	yes	[45]
#103	QD	antihypertensive	*B. mori*	yes	yes	[45]
#104	QE	antihypertensive	*B. mori*	yes	yes	[45]
#105	EG	antihypertensive	*B. mori*	no	yes	[46]
#106	DL	antihypertensive	*B. mori*	no	yes	[46]
#107	GM	antihypertensive	*B. mori*	no	yes	[46]
#108	QK	antihypertensive	*B. mori*	no	yes	[46]
#109	YKPRP	antihypertensive	*G. sigillatus*	no	yes	[47]
#110	PHGAP	antihypertensive	*G. sigillatus*	no	yes	[47]
#111	VGPPQ	antihypertensive	*G. sigillatus*	no	yes	[47]
#112	AFLL	antihypertensive	*G. assimilis*	no	no	[29]
#113	LPLP	antihypertensive	*G. assimilis*	no	no	[29]
#114	DM(+15.99)EKIWH	antihypertensive	*G. assimilis*	no	no	[29]
#115	VFPSIVGRPR	antihypertensive	*G. assimilis*	no	no	[29]
#116	ASTSLEKSY	antihypertensive	*G. assimilis*	no	no	[29]
#117	NILFSGTNVAAGKAR	antihypertensive	*G. assimilis*	no	no	[29]
#118	NPEGLLTGRPR	antihypertensive	*G. assimilis*	no	no	[29]
#119	RYDPNRVF	antihypertensive	*G. assimilis*	no	no	[29]
#120	KPYDLGGRMF	antihypertensive	*G. assimilis*	no	no	[29]
#121	YPLDL	antihypertensive	*G. assimilis*	no	no	[29]
#122	WGPTKPP	antihypertensive	*G. assimilis*	no	no	[29]
#123	FFGT	antihypertensive	*O. smaragdina*	yes	yes	[35]
#124	LSRVP	antihypertensive	*O. smaragdina*	yes	yes	[35]
#125	QGLGY	antihypertensive	*T. molitor*	yes	yes	[48]
#126	NIKY	antihypertensive	*T. molitor*	yes	yes	[48]
#127	HILG	antihypertensive	*T. molitor*	yes	yes	[48]
#128	AVF	antihypertensive	*S. littoralis*	yes	yes	[49]
		antihypertensive	*S. littoralis*	yes	yes	[8]
#129	YAN	antihypertensive	*T. molitor*	yes	no	[50]
		antihypertensive	*T. molitor*	yes	yes	[48]
#130	VF	antidiabetic/antihypertensive	*M. domestica*	no	yes	[28]
		antihypertensive	*S. littoralis*	yes	yes	[8]
#131	QPGR	antidiabetic	*B. mori*	yes	yes	[51] *
#132	SQSPA	antidiabetic	*B. mori*	yes	yes	[51] *
#133	QPPT	antidiabetic	*B. mori*	yes	yes	[51] *
#134	NSPR	antidiabetic	*B. mori*	yes	yes	[51] *
#135	LPPEHDWR	antidiabetic	*B. mori*	yes	yes	[52]
#136	LPAVTIR	antidiabetic	*B. mori*	yes	yes	[52]
#137	APSTIKIKIIAPPER	antidiabetic	*G. assimilis*	no	no	[29]
#138	EITALAPSTIKIK	antidiabetic	*G. assimilis*	no	no	[29]
#139	Q(-17.03)RPEELPLLR	antidiabetic	*G. assimilis*	no	no	[29]
#140	LAMVEA	antidiabetic	*G. assimilis*	no	no	[29]
#141	LPPPP	antidiabetic	*G. assimilis*	no	no	[29]
#142	ALLVVW	antidiabetic	*G. assimilis*	no	no	[29]
#143	DSYPL	antidiabetic	*G. assimilis*	no	no	[29]
#144	EKEEEFENTR	antidiabetic	*G. assimilis*	no	no	[29]
#145	DGMEVPRTP	antidiabetic	*G. assimilis*	no	no	[29]
#146	YPGDV	antidiabetic	*G. assimilis*	no	no	[29]
#147	LPLPL	antidiabetic	*G. assimilis*	no	no	[29]
#148	APVAH	antidiabetic	*T. molitor*	no	no	[9]
#149	AVTTK	antidiabetic	*T. molitor*	no	no	[9]
#150	AAGAPP	antidiabetic	*T. molitor*	no	no	[9]
#151	SLAPK	antidiabetic	*T. molitor*	no	no	[9]
#152	VHCSE	antidiabetic	*T. molitor*	no	no	[9]
#153	PALLL	antidiabetic	*T. molitor*	no	no	[9]
#154	PAALST	antidiabetic	*T. molitor*	no	no	[9]
#155	AR	antidiabetic	*T. molitor*	no	no	[9]
#156	CSR	antidiabetic	*T. molitor*	no	no	[9]
#157	ATAL	antidiabetic	*T. molitor*	no	no	[9]
#158	RVGS	antidiabetic	*T. molitor*	no	no	[9]
#159	AGGP	antidiabetic	*T. molitor*	no	no	[9]
#160	APYF	antidiabetic	*T. molitor*	no	no	[9]
#161	DNKDCFL	antimicrobial	*B. mori*	no	yes	[14]
#162	NNKMNCM	antimicrobial	*B. mori*	no	yes	[14]
#163	TREQWF	antimicrobial	*B. mori*	no	yes	[14]
#164	DNGSGMCK	antimicrobial	*B. mori*	no	yes	[14]
#165	ESCMNCR	antimicrobial	*B. mori*	no	yes	[14]
#166	NDNRINF	antimicrobial	*B. mori*	no	yes	[14]
#167	KDCYTNM	antimicrobial	*B. mori*	no	yes	[14]
#168	SLVDAIGMGP	antithrombotic	*T. molitor*	no	yes	[16]
#169	AGFAGDDAPR	antithrombotic	*T. molitor*	no	yes	[16]
#170	AKKHKE	antioxidant (hepatoprotective)	*T. molitor*	yes	no	[53]
#171	LE	antioxidant (hepatoprotective)	*T. molitor*	yes	no	[53]
#172	PKWF	anti-SARS-CoV-2	*T. molitor*	no	no	[15]
#173	VHRKCF	anti-SARS-CoV-2	*T. molitor*	no	no	[15]
#174	VGVL	hypocholesterolemic	*G. assimilis*	no	no	[29]
#175	PNPNTN	immunomodulatory	*B. mori*	yes	yes	[12]

* Research articles identified through cross-citation search.

## Data Availability

Data is contained within the article or Appendix A.

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
