# Peer review of "Edible Insects as a Novel Source of Bioactive Peptides: A Systematic Review"

_foods, 2023, doi:10.3390/foods12102026_

Round 1

Reviewer 1 Report

This manuscript provides a comprehensive review on the bioactive peptides sourced from edible insects. I found that the manuscript is well focused, and written nicely. I have a very few suggestions as below:

1. Please provide references of the studies (in each row). 

2. Figure 3A has already been represented by Table 1. In fact, Table 1 is more comprehensive (Please see point 1 also) So, no need to provide the same in figure. Figure 3B and 3C are completely okay.

3. Line 474: Please write the scientific name (Mesobuthus martensii) in italics. 

Author Response

This manuscript provides a comprehensive review on the bioactive peptides sourced from edible insects. I found that the manuscript is well focused, and written nicely. I have a very few suggestions as below:

Answer to Reviewer 1: We acknowledge the encouraging comments from reviewer and followed all his suggestions.

  1. Please provide references of the studies (in each row). 

Answer to Reviewer 1: We have inserted the appropriate reference in each row of Table 2 according to the reviewer’s suggestion.

  1. Figure 3A has already been represented by Table 1. In fact, Table 1 is more comprehensive (Please see point 1 also) So, no need to provide the same in figure. Figure 3B and 3C are completely okay.

Answer to Reviewer 1: We have deleted Figure 3A according to the reviewer suggestion.  

  1. Line 474: Please write the scientific name (Mesobuthus martensii) in italics. 

Answer to Reviewer 1: The scientific name was corrected to italic.

Reviewer 2 Report

The manuscript study " Edible insects as a novel source of bioactive peptides: A systematic review". The idea and information provided are interesting. But, this manuscript requires some improvements. Below are some of the technical and non-technical points which should be addressed in order to move on:

Another review article with a similar structure and the following title has been published. What are the innovation and key differences of this study with the mentioned case?

Quah, Y., Tong, S. R., Bojarska, J., Giller, K., Tan, S. A., Ziora, Z. M., ... & Chai, T. T. (2023). Bioactive peptide discovery from edible insects for potential applications in human health and agriculture. Molecules28(3), 1233.

The necessity and innovation of this study are not clear!

It is suggested to state the nutritional characteristics of proteins and peptides obtained from these insects.

Introduction:

The introduction section needs further improvement with the justification of the novelty of the work.

Conclusions: The authors should write this section in comprehensive and suitable style. 

Author Response

The manuscript study " Edible insects as a novel source of bioactive peptides: A systematic review". The idea and information provided are interesting. But, this manuscript requires some improvements. Below are some of the technical and non-technical points which should be addressed in order to move on:

Another review article with a similar structure and the following title has been published. What are the innovation and key differences of this study with the mentioned case?

Quah, Y., Tong, S. R., Bojarska, J., Giller, K., Tan, S. A., Ziora, Z. M., ... & Chai, T. T. (2023). Bioactive peptide discovery from edible insects for potential applications in human health and agriculture. Molecules28(3), 1233.

Answer to Reviewer 2: The referred article is a very interesting and comprehensive review that provides a global overview of the known bioactive peptides derived from insects, as well as the methodologies used to obtain them, their application in human, farm animal and plant health management. However, it is different from our review because this is a systematic review that performs an exhaustive bibliographic search of all research articles reporting sequenced bioactive peptides obtained from edible insects and the respective properties demonstrated by in silico, in vitro and/or in vivo approaches. Moreover, it also discusses the health benefits associated with the consumption of those insects and provides a brief explanation about the mechanism through which the peptides mediate their biological effects. Therefore, both reviews exploit the same issue (bioactive peptides derived from edible insects), but with different purposes. While our review exploits systematically ALL identified bioactive peptides focused on human health, the published review exploits comprehensively the application of SOME bioactive peptides on human health and agriculture. Accordingly, the present review further extends/complements and systematizes the information provided by the published review. Nevertheless, the mentioned review article was inserted in the manuscript (it is reference number 6). The remaining references were correct accordingly.

The necessity and innovation of this study are not clear!

Answer to Reviewer 2: This manuscript is, as far as we know, the first systematic review compiling all information about sequenced bioactive peptides of edible insects, which will be very useful to identify gaps, serving as a starting point for other research works that are much needed to test and validate in vivo the proposed bioactivities.

It is suggested to state the nutritional characteristics of proteins and peptides obtained from these insects.

Answer to Reviewer 2: Information concerning the nutritional characteristics of insect proteins is still scarce, however, an effort was done to include all available and relevant data: lines 57-58 refer that “Insects contain 13-81% of proteins (in dry matter) [2] that are composed of 46-96% of essential amino acids [4], having a digestibility between 76 and 96% [5].” Moreover, when available, we have included the specific nutritional composition of each insect species at the beginning of each sub-section.

Introduction:

The introduction section needs further improvement with the justification of the novelty of the work.

Answer to Reviewer 2: This was emphasized in lines 85-89: As far as we know, this is first the systematic review compiling all information about sequenced bioactive peptides obtained from edible insects. These data will be very useful to identify gaps, serving as a starting point for other research works that are much needed to test and validate in vivo the proposed bioactivities.

Conclusions: The authors should write this section in comprehensive and suitable style.

Answer to Reviewer 2: The conclusions highlight the main findings of this systematic review. Accordingly, they summarize the main bioactive properties identified in peptides of edible insects, which were reported in the literature and, most importantly since this is a systematic review, the main quantitative data are described: number of studies, number of insect species, number of bioactive properties, number of peptides, discriminating those that were submitted to GI digestion and those validated in vivo.

Reviewer 3 Report

The authors have made a thorough selection of publications providing an in-depth evaluation of the potential benefits of insect protein ingestion

Author Response

We acknowledge the reviewer for giving merit to this work.

Reviewer 4 Report

c Title: Edible insects as a novel source of bioactive peptides…. ”

Authors: T.S.S. Teixeira et al.

Obviously the authors have put a lot of effort into this manuscript   - I cannot see anything wrong with their analyses. However, the authors seem to have gleaned over some important aspects that should have received somewhat greater attention and that oversight could be interpreted as if the authors had been uncritically trying to ‘push their own agenda and views’.

They mention global population increase;  fine, but one must not ignore that for half of the world’s population the problem is not a growing population, but a shrinking one! The authors completely ignore that there is no shortage of food whatsoever in western countries, most Asian countries and North America: in the contrary there is a serious overproduction of food  (and the resultant problem of obesity related diseases!

Regarding the global population, the authors are advised to check out: In all European countries and many Asian countries (China recently announced a fall in population and India is expected to reach zero population growth)  populations are decreasing (Population projected to decline in two-thirds of EU-region 2021, available online: https://e.europa.eu/eurostat/web/products-eurostat-news/-/ddn-20210430-2

Nuttall, C.N. 2022. Population decline to the emerging Europe back to the early 20th century. (Available online)

Tsuchiya, H. 2022. East Asia’s looming demographic crisis. https://www.nippon.com/en/in-depth/d00639

Therefore, what needs to be discussed  and to be pointed out is,  why  (if in Asia and Europe populations are decreasing)  should their populations be persuaded to consume insects?  Especially since insects are no better nutritionally than conventional food and especially protein sources. See: Payne, C.L.R.; Scarborough, P.; Rayner, P.; Nonaka, K. “Are edible insects more or less ‘healthy’ than commonly consumed insects? A comparison using two nutrient profiling models developed to combat over- and under-nutrition.”  Eur. J. Clin. Nutr. 2016, 70, 285–291.

See also Ghosh et al. 2017 Journal of Asia-Pacific Entomology, Volume 20, Issue 2, June 2017, Pages 686-694

Actually, insect-based products are not generally increasing in many Asian countries; in the contrary, due to an influx of what could be called western foods the consumption of insects as a food item is on the decline: this holds true for Mexico, NE-India (Arunachal Pradesh), Japan, Thailand and many other countries (see publication by Mueller A 2019:  Insects as food in Laos and Thailand   a case of westernization. Asian J. Soc. Sci. 47, 204-233.).

Of course, the authors are correct and provide valuable data on the insects’ bioactive contents and potential. And, yet again, one must not overlook the antinutrients in some insect species (analysed in and known for some species only: Fig. 7 in Meyer-Rochow et al. 2022 Foods 202110(5), 1036; https://doi.org/10.3390/foods10051036.

Perhaps there are not only ‘benefits’  in using insects as a source of human food  -  in fact what led to the current interest in insects as a food and especially protein source was the 1975 publication by Meyer-Rochow in a journal of the Australian and New Zealand Association of the advancement of Science “Search 6/7, Can insects help to ease the problem of world food shortage? Pp  261-262. That paper which suggested that WHO and FAO support the idea of insects as food for humans should have been mentioned.

Another point is that the authors ignore that Australia (including Papua Niugini), North America (especially Mexico) and even Europe all had a long history of entomophagy (Romans and ancient Greek  feasted on beetle larvae and other insects).  What I always find surprising is that authors tend to ignore that Mexico is part of North America and Mexico did have a  tradition of consuming insects  (although the consumption of insects in Mexico is nowadays in severe decline).

Finally, it makes little sense, in fact it is misleading, to show photographs of 13 insects (scorpions are NOT insects at all!!!), which are not  consumed as adults, but as larvae only. And therefore, the question: are the authors analysing bioactive compounds of the larvae or the adults?  Would there be differences? I’d definitely think so.

Author Response

Obviously the authors have put a lot of effort into this manuscript   - I cannot see anything wrong with their analyses. However, the authors seem to have gleaned over some important aspects that should have received somewhat greater attention and that oversight could be interpreted as if the authors had been uncritically trying to ‘push their own agenda and views’.

Answer to Reviewer 4:  This is a systematic review on bioactive peptides of insects, not a comprehensive review on insects as food, therefore reviewer comment that authors had been uncritically trying to ‘push their own agenda and views’ is not understandable. All statements are based on existing literature and data from the United Nations (UN).

They mention global population increase;  fine, but one must not ignore that for half of the world’s population the problem is not a growing population, but a shrinking one!

Answer to Reviewer 4: We based our discussion on the UN data and projections https://www.un.org/en/global-issues/population for the WORLD population. Local demography is outside the scope of this systematic review.

The authors completely ignore that there is no shortage of food whatsoever in western countries, most Asian countries and North America: in the contrary there is a serious overproduction of food  (and the resultant problem of obesity related diseases!

Answer to Reviewer 4: We do not understand why this question is raised because we did not ignore or include any particular data, but instead referred globally. It is mentioned in the manuscript that this is  a future problem, not an actual problem: “According to the United Nations projections, the world’s population is expected to grow from the current 8 billion in 2022 to nearly 9.7 billion in 2050 (https://www.un.org/en/global-issues/population), which will demand a dramatic intensification of food and feed production. Additionally, the decrease of cultivation areas resulting from the climate changes and industrial development, together with the effects of the temperature changes on the crop yields are serious challenges to overcome by the next generations.” (Lines 28-34).

Regarding the global population, the authors are advised to check out: In all European countries and many Asian countries (China recently announced a fall in population and India is expected to reach zero population growth)  populations are decreasing (Population projected to decline in two-thirds of EU-region 2021, available online: https://e.europa.eu/eurostat/web/products-eurostat-news/-/ddn-20210430-2

Nuttall, C.N. 2022. Population decline to the emerging Europe back to the early 20th century. (Available online)

Tsuchiya, H. 2022. East Asia’s looming demographic crisis. https://www.nippon.com/en/in-depth/d00639

Therefore, what needs to be discussed  and to be pointed out is,  why  (if in Asia and Europe populations are decreasing)  should their populations be persuaded to consume insects?  Especially since insects are no better nutritionally than conventional food and especially protein sources. See: Payne, C.L.R.; Scarborough, P.; Rayner, P.; Nonaka, K. “Are edible insects more or less ‘healthy’ than commonly consumed insects? A comparison using two nutrient profiling models developed to combat over- and under-nutrition.”  Eur. J. Clin. Nutr. 2016, 70, 285–291.

See also Ghosh et al. 2017 Journal of Asia-Pacific Entomology, Volume 20, Issue 2, June 2017, Pages 686-694

Answer to Reviewer 4: This systematic review does not say that insects are nutritionally better than any other food, instead environmental and economic advantages are mentioned: Their breeding has several environmental and economic advantages compared to the traditional protein sources (meat and plants), including: (i) high reproduction rate, (ii) high feed conversion efficiency, (iii) small areas of rearing land required, avoiding deforestation, (iv) minor water needs, (v) lower emission of greenhouse gases and ammonia, (vi) lower economical investment in technology, and (vii) potential to reduce the use of insecticides when the collected insects are considered pests (e.g. desert locust) [4].” (Line 49-56)

Actually, insect-based products are not generally increasing in many Asian countries; in the contrary, due to an influx of what could be called western foods the consumption of insects as a food item is on the decline: this holds true for Mexico, NE-India (Arunachal Pradesh), Japan, Thailand and many other countries (see publication by Mueller A 2019:  Insects as food in Laos and Thailand   a case of westernization. Asian J. Soc. Sci. 47, 204-233.).

Answer to Reviewer 4: We do not understand why this question is raised because we did not say any of the referred comments. The data provided by the reviewer corroborate our opinion that new scientific studies are needed to convince the consumers to included insect-based products in their diets.

Of course, the authors are correct and provide valuable data on the insects’ bioactive contents and potential. And, yet again, one must not overlook the antinutrients in some insect species (analysed in and known for some species only: Fig. 7 in Meyer-Rochow et al. 2022 Foods 2021, 10(5), 1036; https://doi.org/10.3390/foods10051036.

Answer to Reviewer 4: We included that information and the respective reference in the manuscript: ”Despite all the environmental, economic, and nutritional advantages associated with the introduction of insects in human diet, there are some health risks that demand their careful assessment, such their anti-nutrient contents [17] and the possibility of causing adverse allergic reactions [18]. “ (Line 71)

Perhaps there are not only ‘benefits’  in using insects as a source of human food  -  in fact what led to the current interest in insects as a food and especially protein source was the 1975 publication by Meyer-Rochow in a journal of the Australian and New Zealand Association of the advancement of Science “Search 6/7, Can insects help to ease the problem of world food shortage? Pp  261-262. That paper which suggested that WHO and FAO support the idea of insects as food for humans should have been mentioned.

Answer to Reviewer 4: We added the suggested reference and a new sentence in lines 37-39.

Another point is that the authors ignore that Australia (including Papua Niugini), North America (especially Mexico) and even Europe all had a long history of entomophagy (Romans and ancient Greek  feasted on beetle larvae and other insects).  What I always find surprising is that authors tend to ignore that Mexico is part of North America and Mexico did have a  tradition of consuming insects  (although the consumption of insects in Mexico is nowadays in severe decline).

Answer to Reviewer 4: To avoid ignoring any region, we changed the sentence to: “Presently, it is estimated that 2086 insect species are consumed as foods in 3071 ethnic groups from 130 countries [2].” (Lines 40-42).

Finally, it makes little sense, in fact it is misleading, to show photographs of 13 insects (scorpions are NOT insects at all!!!), which are not consumed as adults, but as larvae only. And therefore, the question: are the authors analysing bioactive compounds of the larvae or the adults?  Would there be differences? I’d definitely think so.

Answer to Reviewer 4: We have removed all data on the Buthus martensii (scorpion) from the manuscript and from all figures. Accordingly, we have done all the necessary changes along manuscript. As mentioned in the manuscript, this systematic review compiles all information on sequenced bioactive peptides obtained from edible insects. The life cycle phase of each insect is not an exclusion/inclusion factor and was already described in the manuscript text. Figure 2 was replaced by a new one, containing the 12 insect species at the exact stage of their life cycle referred to in the articles from which the information (relating to the peptides) was taken.

Round 2

Reviewer 4 Report

The value of this article ms is that it focuses on bioactive peptide. It therefore provides useful information for nutritionists and people interested in using insects and insect compounds therapeutically.

Although I still feel you authors should have addressed some of the points regarding global populations and assumed food shortages, let's not old up publication of this interesting article because of that issue.  However, I urge you, the authors, to keep my earlier comments in mind when you follow up this research with some more publications.

The only small thing I'd like the authors to do now, is related to Table 1, in which they list the orders of insects as "Coleoptera, Orthoptera, Hymenoptera, Lepidoptera, Diptera".  Orthoptera, as an order of hemimetabolous insects, should be listed before, i.e. above 'Coleoptera' and the remaining orders, which are all holometabolous insects.  If that can NOT be done, then the authors are advised to explain in the table legend, why they used a sequence of the insect orders in which Orthoptera follow Coleoptera, which is unusual and must have a reason.

Author Response

Answer to reviewer 4: As suggested, the lines of Table 1 were reordered and the Orthoptera species were listed before all other species. The subchapters 3.1.1 to 3.1.12 and the photographs of the species in the Figure 2 were also reordered accordingly.